# Metric information in cognitive maps: Euclidean embedding of non-Euclidean environments

**Tristan Baumann** [1] *, **Hanspeter A. Mallot** [2]

**1** Computational Neuroscience, Max Planck Institute for Biological Cybernetics, Tübingen, Germany,
**2** Cognitive Neuroscience Unit, Department of Biology, University of Tübingen, Tübingen, Germany

* tristan.baumann@tuebingen.mpg.de

## Abstract

The structure of the internal representation of surrounding space, the so-called *cognitive map*, has long been debated. A Euclidean metric map is the most straight-forward hypothesis, but human navigation has been shown to systematically deviate from the Euclidean ground truth. Vector navigation based on non-metric models can better explain the observed behavior, but also discards useful geometric properties such as fast shortcut estimation and cue integration.

Here, we propose another alternative, a Euclidean metric map that is systematically distorted to account for the observed behavior. The map is found by embedding the non-metric model, a labeled graph, into 2D Euclidean coordinates. We compared these two models using data from a human behavioral study where participants had to learn and navigate a non-Euclidean maze (i.e., with wormholes) and perform direct shortcuts between different locations. Even though the Euclidean embedding cannot correctly represent the non-Euclidean environment, both models predicted the data equally well. We argue that the embedding naturally arises from integrating the local position information into a metric framework, which makes the model more powerful and robust than the non-metric alternative. It may therefore be a better model for the human cognitive map.

## Author summary

How is the metric of space, i.e., knowledge about distances and angles between places, represented in the brain? Existing theories argue for either purely relational topological graphs without a metric, or consistent Euclidean maps where each place is assigned specific coordinates. The problem lies in the fact that human behavior systematically deviates from perfect metric maps, and theories need to account for these deviations.

We propose an intermediate model that has both properties of non-metric graphs and metric maps, by embedding a graph labeled with local position information into metric space. In this "embedded graph", measurements of local metric information also affect the estimates of adjacent distances and turning angles. The result is a consolidated spatial representation which is still a graph, but whose local metric labels are globally optimized

**Data Availability Statement:** Code and data are available under DOI: 10.5281/zenodo.8158597.

**Funding:** The author(s) received no specific funding for this work.;

**Competing interests:** The authors have declared that no competing interests exist.

to match the available egomotion measurements. We show that the embedded graph is consistent with human behavior in a (virtual) non-Euclidean environment and argue that it is a natural consequence of the optimal integration of repeated local measurements over time.

## Introduction

### The cognitive map

The spatial long-term memory contains representations of places, landmarks, and local views. A sequence of navigational actions connecting these representations is called a route and animals with such route knowledge are able to navigate between known places by following these routes [1–4]. If knowledge about many different items, places, and routes is integrated and novel routes and shortcuts can be inferred from previously learned route segments, the representation is called a map [4–8]. The cognitive map is thus a form of declarative memory in the sense that it characterizes "knowing what" or "knowing where" as opposed to the non-declarative "knowing how" of routes or guidance information [6, 9].

A cognitive map is the most general form of spatial long-term memory, and it is believed that many animals, including humans, have access to this representation [3, 7, 10]. This is exemplified by the existence of neural correlates of position, the place cells [6, 11–13], which encode the position of the animal within the current context via population activity.

The intuition of an internal map is relatively straight-forward, because it matches maps encountered in everyday life: In general, such maps may be broadly characterized by two frameworks: Euclidean metric maps and topological graphs. Euclidean metric maps, such as a bird's eye view of a city or a satellite image, assign unique coordinates to each position that approximate the real-world geometry by preserving the metric relationships between positions. Topological graphs, such as a subway or bus chart or an instruction manual, describe states and possible actions that lead from one state to another, rather than geometry.

The metric framework (Fig 1c) is considerably better suited to explain environments with a Euclidean geometric structure, and, based on the Kantian notion of an *a priori* assumption of absolute external space [14], it has often been argued that the cognitive map must likewise follow the laws of the Euclidean metric to capture these properties [6, 7, 10, 15]. This theory is supported by the existence of grid cells in the entorhinal cortex, which are believed to encode metric path integration information [15–17].

The notion of an absolute Euclidean metric may be challenged, e.g., by pointing out that the intuition of straight lines on a curved surface (or any surface that is not a plane) are actually geodesics and not true straight lines in an Euclidean sense [4, 18, 19]. But even an approximately Euclidean or non-Euclidean metric map may be advantageous, since geometric relationships between places are preserved in a highly efficient manner. That is, distances, routes, and shortcuts can be directly inferred from the map and need not be memorized individually. This property enables metric maps to store an immense amount of data, making them powerful informational tools [10].

However, results from navigation experiments often disagree with the Euclidean metric map hypothesis: Human performance in shortcutting or triangle completion from long-term memory, which have been taken as evidence for an Euclidean representation, is highly unreliable with angular errors of over ±90° and angular standard deviations between 25° − 45° [3, 20–22]. The Euclidean metric postulates are often violated and angle and distance estimations are systematically biased by features of the environment such as landmarks, junctions or

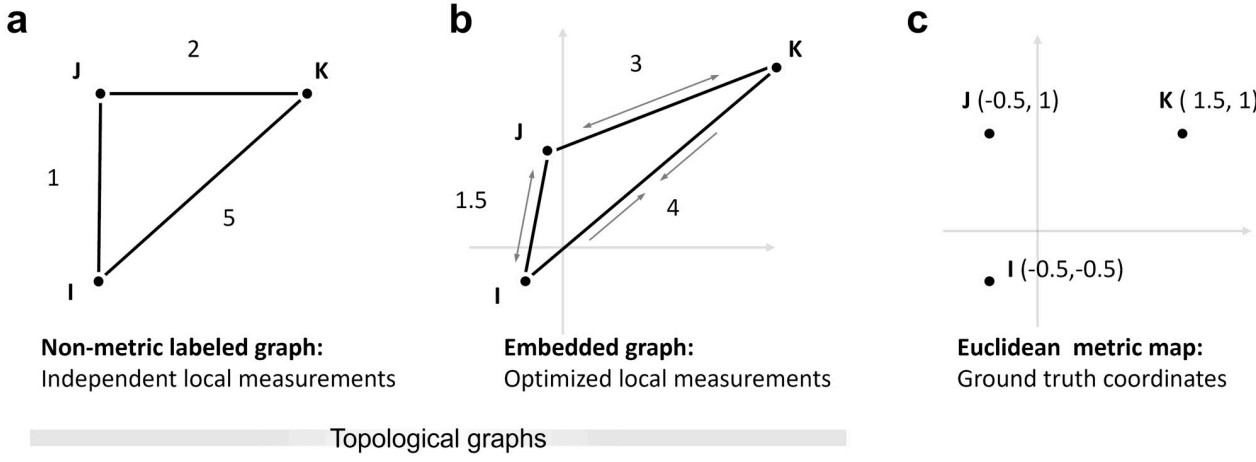

**Fig 1. Cognitive map hypotheses. (a)** Non-metric topological graph, labeled with distance measurements. The labels are independent of each other and do not need to adhere to the triangle inequality. **(b)** Embedded graph from (a). To find a Euclidean embedding, the distance labels need to be adjusted to create a valid configuration, for example by stretching or compressing the edges or "wiggling" on the vertices until the difference between map and labels is minimized. As opposed to the non-metric labeled graph, changes to one label will therefore influence others. **(c)** Euclidean metric map. Places are directly assigned coordinates based on their position in the world. Over time, the coordinates may be refined by repeated measurements and the map will approach the Euclidean ground truth. The same can be expected from the embedded graph optimization if the labels are refined.

region boundaries [3, 23–27], or the number and recency of preceding turns [17, 28, 29]. In rats, place cells have been shown to stretch and shear following room deformation, while still preserving topological information about the environment [30, 31]. These results imply that space is encoded with lower fidelity than what a precise metric map would predict.

As an alternative, the comparatively weaker class of topological graphs is often proposed. The environment is expressed through neighborhood or adjacency relations, forming a network of places as graph vertices and paths or actions connecting them as edges [3, 17, 32–34]. The graph may be labeled with pairwise distance or angle measurements, but this information need not adhere to the metric postulates and is therefore not metric (Fig 1a). Still, shortcuts and novel routes can be derived via vector addition of the labels along paths in the graph; indeed, Warren et al. (2017) [35] suggest that vector addition based on labeled graphs best explains human performance in navigation experiments.

Nevertheless, poor navigational performance, biases, and large errors are not enough to completely rule out a Euclidean metric representation because the map may be systematically biased or distorted to large degrees while still being metric [3]. Overall, the errors of inferred distances in distorted metric maps are likely smaller than in non-metric labeled graphs, where distance labels are independent. A means to find such distorted metric representations by efficiently exploiting all available distance information, is *metric embedding*.

## Distorted maps and non-Euclidean environments

Each individual cognitive representation will generally be different due to acquisition order, biases, and accumulation of measurement errors. One possible advantage of map-like representations in spatial memory is the mutual refinement of (possibly conflicting) local position information over time: As the agent explores the environment, it will repeatedly obtain distance and angle measurements of connections between the known places or graph states.

With a topological graph, repeated measurements of the same information could be used to create more precise labels by averaging. However, the labels will always remain independent of labels corresponding to adjacent connections and, in a triangle, might persistently violate the triangle inequality which defines a mathematical metric (Fig 1a). In the following, we therefore refer to this representation as the *non-metric labeled graph.*

Additional precision can only be gained if repeated measurements of one connection will also improve estimates along other connections in the graph. This may for example be achieved by *metric embedding* (Fig 1b). Metric embedding is a means of finding a representation of the non-metric labeled graph in 2D or 3D Euclidean space in a way that best reproduces the spatial information contained in the graph. Since the acquisition of spatial memory is not complete after a single pass through the environment but relies on the consolidation of many local measurements, metric embedding seems to be a natural method for continuous integration of local information. In this sense, cue integration might be the main reason for organizing spatial representations in a metric framework.

If the measured labels are not perfect, Euclidean metric embedding can only approximate the true Euclidean metric relations and will result in a distorted representation. The so *embedded graph* could therefore be an alternative metric explanation for the large deviations in human navigation, as opposed to the non-metric labeled graph.

In regular environments, differences between a non-metric labeled graph, an embedded graph, and a Euclidean metric map will be minimal, because the models are likely to approach the same underlying ground truth as measurements are refined. Therefore, cases need to be considered in which the models would make different predictions. With the advent of immersive virtual reality, a unique opportunity has opened up to present non-Euclidean environments, thus dissociating presented metric information from the underlying true Euclidean positions [35–38]. The non-Euclidean manipulations have been shown to heavily influence navigation but are usually not noticed by the subjects [35, 36].

## Evidence from wormhole experiments

In the following, we focus on a specific example, Warren et al. (2017) [35], because the experiment offers an excellent setup to investigate the hypotheses with respect to systematic distortion and the data are available online.

Warren et al. (2017) [35] presented participants with a non-Euclidean environment and argued that, if the cognitive map has a Euclidean metric, participants should have greater difficulties in learning the non-Euclidean environment compared to control, because mismatches between the cognitive map and the environment should occur. On the other hand, a non-metric graph should have no such issues.

Using head-mounted display virtual reality, Warren et al. created a hedge maze augmented with two invisible wormholes. The wormholes functioned as instant seamless teleportation and 90˚ rotation between different parts of the maze while participants continued to walk normally in the real-life room, therefore creating a mismatch between maze position and path integration information. Interestingly, only one out of fifty participants reported noticing any kind of spatial anomaly in the maze.

Participants had to memorize object positions within the maze and were later asked to walk direct shortcuts between them. For this, the participants were moved to a starting object and had time to orient themselves. Then, the walls of the maze disappeared, and the participants had to walk to the presumed position of a target object. The initial angles of the subjects' trajectories were measured and used as directional estimates to compare the non-metric labeled graph and undistorted Euclidean map models.

Warren et al. found that directional estimates were heavily distorted towards the worm-holes. This is predicted by vector addition along the shortest path on a labeled graph but not by straight lines in Euclidean ground truth coordinates. The authors thus rejected the Euclidean map hypothesis in favor of the non-metric labeled graph, arguing that only a non-Euclidean structure could explain the observed results [3, 35]. A distorted Euclidean map was briefly considered but rejected on the basis that such a map "must still satisfy the metric postulates [. . .] in the inertial coordinate system" [35]. However, the metric embedding is not a simple integration of local position information (Warren's "inertial coordinates") but the mutual consolidation of distance and angle information over the entire graph. I.e., two different positions in undistorted ground truth coordinates may very well occupy the same position in the distorted embedding and vice-versa. Therefore, deviations from the ground truth do not imply that the representation is not Euclidean, but only that the representation does not match the ground truth.

We reexamined the data reported in Warren et al. (2017) [35] with respect to the possibility of a distorted Euclidean map. In the following, we show that such a map can be found by first creating a non-metric labeled graph for the maze and then embedding the graph into 2D Euclidean coordinates. This is achieved by the minimization of the angle and distance differences between graph and map, following the method described in Hübner and Mallot (2007) [39] and Mallot (2024) [4] for the embedding of view graphs. In an ordinary Euclidean environment, the embedding will recover the ground truth coordinates, but in a non-Euclidean environment, a residual error between embedding and local measurements must remain. Because of this error, the models should make different predictions, and may be distinguished by comparing their predictions to experimental data. That is, shortcuts derived from the embedded graph should fall somewhere between the shortcuts from the other two models.

However, we found that both models, the non-metric labeled graph and its Euclidean metric embedding, were able to predict the data equally well. Because the embedded graph is a valid Euclidean map, it is better suited for shortcut generation and especially cue integration than the non-metric alternative. We therefore refute the claim by Warren et al. (2017) [35] that their findings cannot be explained by a Euclidean metric map and argue for the embedded graph as a better alternative explanation.

## Materials and methods

### Data acquisition

The data used here are figures, measurements, and results from Warren et al. (2017) [35]. The anonymized per-subject measurements are available as supplementary material online in the Brown University Digital Repository (http://dx.doi.org/10.7301/Z0JS9NC5, retrieved in November 2022). The relevant datasets contain measurements of the direction of individual shortcuts between object pairs in the wormhole maze, given as angular difference between the estimate and the straight-line direction in Euclidean ground truth coordinates. We estimated these coordinates from pixel positions based on Fig 2B in Warren et al. (2017) [35] (Fig 2a) and transformed the subject estimates into global angles (i.e., increasing counterclockwise from the positive x-axis or east). The layout of the maze and example subject estimates are shown in Fig 3.

Warren et al. (2017) [35] measured direction estimates in two separate experiments, one to investigate shortcuts (Dataset "Route-finding and shortcuts", see Fig 3b) and one to investigate the ordinal reversal of landmark positions (Dataset "Rips and folds", see Fig 3c). "Route-finding and shortcuts" contains directional estimates of 10 subjects (5M, 5F) for four pairs of objects for a total of $10 \times 4 \times 2$ (bidirectional) = 80 measurements. "Rips and folds" contains

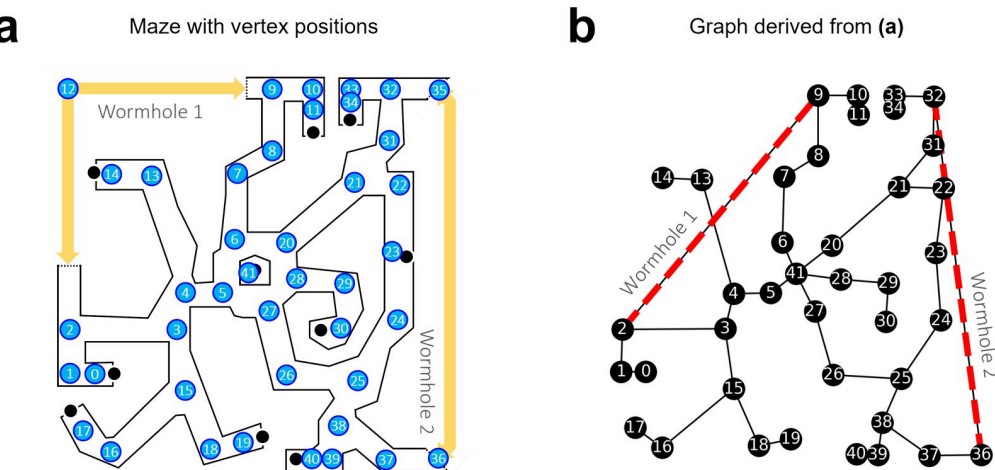

**Fig 2. Graph creation. (a)** Vertex positions in the maze. Their pixel coordinates were considered the Euclidean ground truth for the model. The maze was partitioned into straight segments and corners, and one vertex was placed per corner. Two vertices, 12 and 35, were only used in a control graph without wormholes. **(b)** The corresponding topological graph with edges through wormholes (red dotted lines). The graph was then labeled with local distance and angle measurements based on the ground truth, except for the wormhole edges, which were manually adjusted to reflect the locally distorted topology instead. Note that the distance along the wormhole edges is shortened but not zero.

directional estimates of 11 subjects (9M, 2F) for eight starting locations and three targets each for a total of $11 \times 8 \times 3 = 264$ measurements. For the purpose of this study, both datasets were treated in the same way but were evaluated separately; this was done for direct comparison and to avoid bias because some subjects may have participated in both studies. For further information about the participants, hardware, and experimental setup please refer to the original paper by Warren et al. (2017) [35].

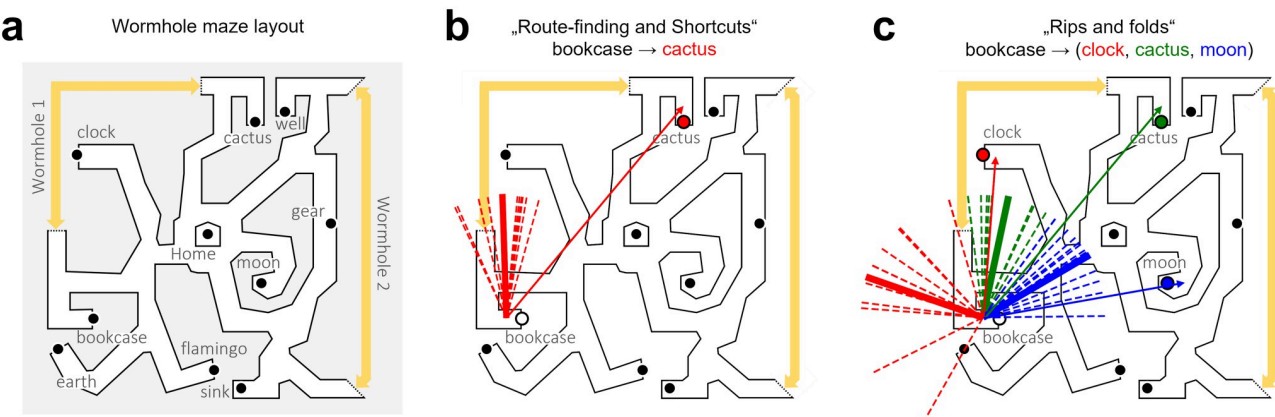

**Fig 3. Maze and shortcut data. (a)** Layout of the wormhole maze, redrawn from Warren et al. (2017) [35]. The yellow arrows show wormhole position and magnitude. Touching one end of the arrow instantly and seamlessly teleported subjects to the other end. **(b, c)** Example directional estimates for object pairs from the "Route-finding and shortcuts" dataset (Experiment 1 in Warren et al. (2017) [35]) and the "Rips and folds" dataset (Experiment 2 in Warren et al. (2017) [35]). The thin arrows show the Euclidean ground truth direction between objects, the short dotted lines the corresponding subject estimates, and the thick solid line the average subject estimate. The length of the estimates has been normalized and does not reflect walked distance. In (c), the colors indicate different goals.

## Graph and map setup

Plausible Euclidean embeddings were found in two steps: First, a topological graph of the maze was created and labeled with the veridical distances and angles. This graph was also used to derive predictions for the non-metric labeled graph hypothesis, i.e., vector addition of the labels along the shortest paths. Next, the graph was embedded into 2D Euclidean coordinates by iterative minimization of a stress function [4, 39] describing the difference between the coordinates and the local labels.

In general, the creation of the topological graph is a non-trivial problem with a possibly infinite set of solutions consisting of any number of vertices, edges and measurements along the maze. Therefore, good solutions have to be guessed. We used the ground truth of the wormhole maze as a basis (Fig 2a); due to its well-defined straight segments and corners, it can easily be reduced to a simplified graph by placing one vertex per corner and one edge per straight segment (Fig 2b). Formally, we define the graph $G = \{V, E\}$ as a set of $n$ vertices $V = \{v_1, \ldots, v_n\}$ corresponding to places in the maze and edges $E = \{e_{ij}, e_{jk}, \ldots\}$ describing maze arms connecting the places $v_i$ to $v_j$ and $v_j$ to $v_k$.

The algorithm for metric embedding is based on local distance and turning information only, without the assumption of a global reference direction (e.g., north). It is therefore based on triplets of neighboring places $T = \{(i, j, k)\}$, i.e., places that can be visited in sequence. For each triplet, the local distances $d_{ij}$, $d_{jk}$ and the turning angle $\alpha_{ijk}$ were measured in the ground truth maze and added as labels to the topological graph. $d_{ij}$ and $d_{jk}$ describe the distances between places $i$, $j$ and $j$, $k$ and $\alpha_{ijk}$ the heading change at $j$ when moving from $i$ to $k$. All labels were taken from the required egomotion steps such that labels around wormholes differed from the Euclidean ones. I.e., the labeled graph perfectly matches the local geometry encountered throughout the maze, including the passage through wormholes. The same labeled graph was used for datasets from all subjects.

From the graph, a 2D Euclidean embedding $X = \{(\mathbf{x}_1, \ldots, \mathbf{x}_n)\}$ of the $n$ vertices was derived by minimizing the following stress function: The algorithm considers all measured triplets of neighboring places $T = \{(i, j, k)\}$ and their related distance and angle measurements ($d_{ij}$, $d_{jk}$, $\alpha_{ijk}$). Each place may appear many times as part of different triplets, and forward-backward movements of the form $(i, j, i)$ are also considered (with $\alpha_{ijk} = 180°$). The stress function can then be written as

$$f(\mathbf{x}_1, \ldots, \mathbf{x}_n) =$$
$$\sum_{(i,j,k) \in T} \lambda_1 [((\mathbf{x}_j - \mathbf{x}_i) \cdot (\mathbf{x}_j - \mathbf{x}_k)) - d_{ij} d_{jk} \cos\alpha_{ijk}]^2 + \tag{1}$$
$$\lambda_2 [((\mathbf{x}_j - \mathbf{x}_i) \otimes (\mathbf{x}_j - \mathbf{x}_k)) - d_{ij} d_{jk} \sin\alpha_{ijk}]^2.$$

here, $(\cdot)$ denotes the dot product and $(\otimes)$ the third component of the cross product, $(\mathbf{a} \otimes \mathbf{b}) := a_1 b_2 - a_2 b_1$, which is twice the area of the triangle $(i, j, k)$. The constants $\lambda_1$, $\lambda_2$ can be used to weigh the components based on their variances [4]; we chose $\lambda_1 = \lambda_2 = 1$.

Finding an embedding that minimizes this stress function is a nonlinear optimization problem. Solutions may for example be found with iterative numerical approximations like Newton's method. We used the quasi-Newton method Sequential Least Squares Programming (SLSQP), as implemented in the *SciPy 1.10 optimize* Python library [40], credited to [41].

The resulting embedding will be a Euclidean metric map of the graph's vertices with an arbitrary global orientation, but it is not a complete distorted map of the wormhole maze in the sense that it only assigns coordinates to the vertices but not to arbitrary places. The distorted position of other places may be found by adding them as additional vertices to the

graph before embedding or by interpolation. Nevertheless, the embedding is sufficient to derive directional predictions.

## Model comparison and data analysis

Next, the non-metric labeled graph and its Euclidean embedding were used to derive predictions about shortcut directions between object pairs. For the non-metric labeled graph, predictions were obtained by finding the shortest path between start and target object using Dijkstra's algorithm, as implemented in the *NetworkX 3.0* Python library [42]. Along the path, the angles and distances were summed up to a vector, and the global direction of the resultant vector relative to the ground truth coordinates was considered the final shortcut prediction. In the embedded graph, shortcuts were simply the straight lines from start to target objects.

The predictions of the two graph models were compared to the subject data and the prediction error was measured. Because the embedded graph has no defined reference direction, subject estimates had to be considered relative to a local reference. We used the respective local angle between the starting arm and measurement or prediction, which is independent of the reference direction.

For each model, the mean prediction errors and between-subject angular deviation were calculated for the group, and the within-subject angular deviation separately for each participant. The errors were compared with the two-sample Watson-Williams F-test for circular data [43], as implemented in the *PyCircStat* Python library [44]. The null hypothesis assumes that the samples come from underlying distributions with the same mean [43], i.e., that the models explain the subject data equally well; note that this does not mean that the models make the same prediction. Cohen's $d$ was used as a measure for effect size. All statistical tests were two-tailed with $\alpha = 0.05$.

We then compared the models using the Bayesian information criterion (BIC) [45]. The BIC is based on the maximum likelihood of observing the data given a specific model and penalizes the number of free parameters in the model. Generally, a model with a lower BIC is preferable.

To obtain likelihood functions for the prediction errors, we added a noise term describing the inter-subject variation to both models. The noise was modeled as a von Mises distribution using the empirical prediction error means and variances. In the embedded graph, the free parameters are the $n \times 2$ coordinates of the 2D Euclidean embedding $X$ and a noise term for a total of 82 free parameters. To fully specify the non-metric labeled graph model, the required parameters are the set of all distance labels $d$ and angle labels $\alpha$, as well as the noise term for a total of 162 parameters. Note that these definitions are only valid in the respective Euclidean- and graph-based frameworks which, for example, come with different distance functions (straight lines in the Euclidean framework and the shortest path in the graph-based framework).

## Results

### Embeddings

The numerical optimization method may find different local minima. Which solution is found depends on the starting point in the solution space, i.e., the initial vertex positions $X = \{(\mathbf{x}_1, \ldots, \mathbf{x}_n)\}$. We restarted the optimization procedure 1000 times with random initial vertex positions $X \sim \mathcal{U}_2(0, 20)$ and found two local minima with stress values $f(X_1) = 450.68$ (Fig 4c) and $f(X_2) = 367.36$ (Fig 4b). In the following, we report results from the first embedding, which resulted in better fits to the subject data.

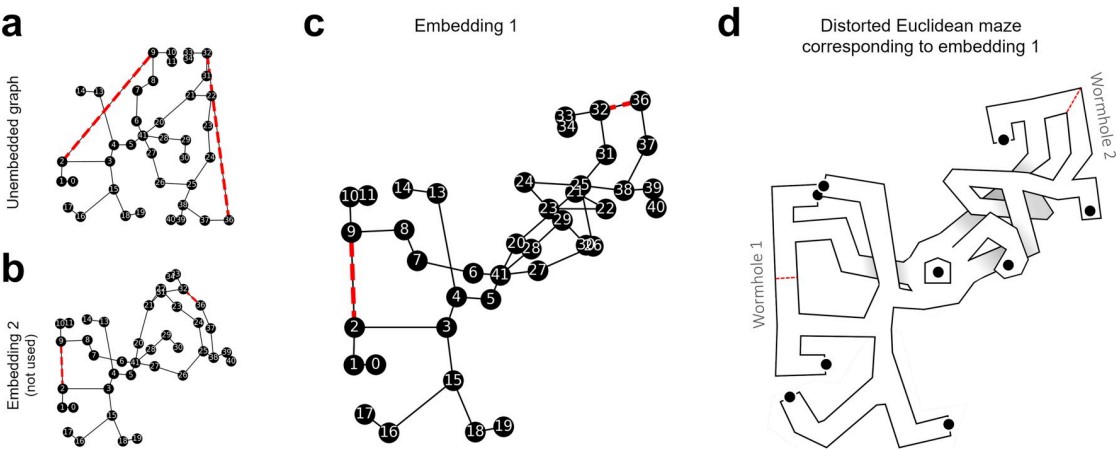

**Fig 4. Embedded graph. (a, b)** References for comparison. The unembedded graph (as in Fig 2) and an another embedding which was also found by the optimization method. The second embedding performed worse on the subject data and was not further used. **(b)** The embedded graph, i.e., the labeled graph with the vertices at coordinates that minimize the difference between map and labels. The orientation of the embeddings is arbitrary; here, they were rotated so that the edge (2, 3) is horizontal. The red dotted lines show the edges that pass through wormholes. **(d)** Sketch of the distorted wormhole maze according to the embedding in (c). The sketch shows how the embedding might be represented by a subject. Edges that cross each other in the embedded graph could for example be rationalized as multi-level paths, leading to a 3D representation. Alternatively, in a purely 2D map, the arms would simply intersect. Note that the edges have no coordinates in the embedding but are simply lines in the adjacency matrix.

## Dataset 1: Route-finding and shortcuts

We derived shortcut predictions from the non-metric labeled and embedded graph models and compared the predictions to human shortcut estimates from Warren et al. (2017) [35], dataset "Route-finding and shortcuts" (Fig 5a–5c). The resulting angular prediction error was measured. Rayleigh tests on error direction revealed non-uniform distributions, $z(10) = 9.59$, $p < .001$ for the non-metric labeled graph and $z(10) = 9.57$, $p < .001$ for the embedding.

The non-metric labeled graph model showed an average angular error of $-12.4°$ with an angular deviation ($AD$) of $11.76°$ and the embedding an average error of $-15.26°$, $AD = 11.98°$. This difference was not significant ($F(1, 18) = 0.2$, $p = .63$) with a small effect size ($d = .22$). I.e., the shortcut directions derived from the graph model were not significantly closer to the subject data than the shortcut directions derived from the embedding or vice versa.

The within-subject angular deviation of the errors was fairly high but also similar for both models, with an average of $29.75°$ for the graph model and $32.15°$ for the embedding. Statistical comparison ($F(1, 18) = 0.6$, $p = .42$, $d = .51$) again revealed no significant difference. Given the similar prediction errors but large difference in free parameters, the Bayesian information criterion strongly favored the embedding over the non-metric labeled graph ($BIC_{embedding} = 221.7$ vs. $BIC_{non\text{-}metric} = 405.97$).

## Dataset 2: Rips and folds

For the purpose of this study, the "Rips and folds" dataset was treated the same as the "Route-finding and shortcuts" dataset, with the only difference being the number of participants (11 vs. 10 in dataset 1) and estimates per participant (24 vs. 8 in dataset 1). The datasets were analyzed separately for the sake of comparison.

We again compared prediction errors of the non-metric labeled graph model and its Euclidean embedding (Fig 5d–5f). Rayleigh test on error direction revealed non-uniform distributions, $z(11) = 10.78$, $p < .001$ for the non-metric labeled graph and $z(11) = 10.77$, $p < .001$

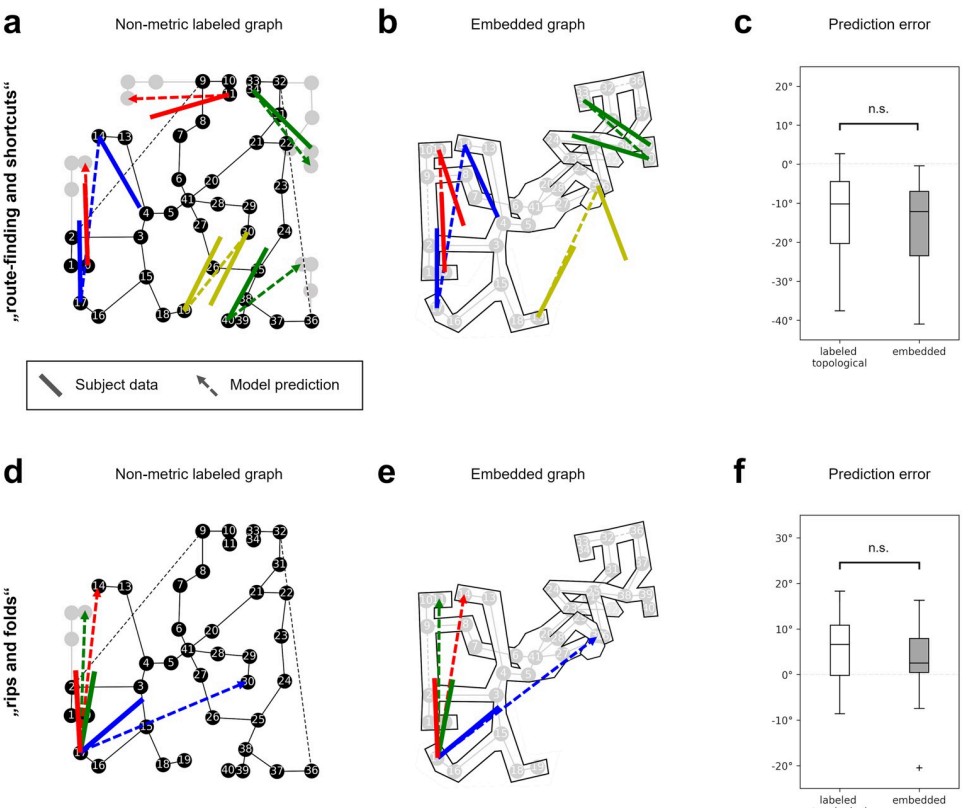

**Fig 5. Results. (a-c): Dataset "route-finding and shortcuts", (d-f): Dataset "rips and folds".** **(a)** Shortcut predictions of the non-metric labeled graph (dotted lines) and average subject estimates (solid lines), plotted on ground truth coordinates. The gray vertices show how the graph would continue on routes through wormholes. **(b)** Shortcut predictions of the embedded graph, lines as in (a). The subject estimates were rotated to match the local orientation of the originating maze arm. **(c)** Distribution of the prediction error. The difference between the models is not significant, i.e., they predict the data equally well. **(d)** Example shortcut predictions (dotted lines) and subject estimates (solid lines) for three of the 24 object pairs in the "rips and folds" dataset. **(e)** Shortcut predictions of the embedded graph for the same object pairs as in (d). **(f)** Distribution of the prediction error. The difference between the models is also not significant on this dataset.

for the embedded graph. The non-metric labeled graph showed an average angular error of 5.68˚, $AD$ = 8.12˚, and the embedded graph an error of 2.37˚, $AD$ = 9.35˚. This difference was again not significant ($F(1, 20)$ = 0.72, $p$ = .41) with a small effect size ($d$ = .39). Within-subject angular deviation of the errors was also high, with an average of 42.36˚ for the non-metric labeled graph model and 33.77˚ for the embedding. This difference was trending towards significance ($F(1, 20)$ = 4.07, $p$ = .057) with a large effect ($d$ = .85).

Although dataset 2 contained many more measurements than dataset 1, there was still no significant difference between the prediction errors, i.e., the models again predicted the data equally well. As before, the Bayesian information criterion strongly favored the Euclidean embedding over the non-metric alternative ($BIC_{embedding}$ = 233.66 vs. $BIC_{non-metric}$ = 425.92).

## Discussion

Using subject data from Warren et al. (2017) [35], we compared two cognitive map models, the non-metric labeled graph and the embedded graph. We found both models predicted the

data equally well, i.e., both models made prediction errors with a similar magnitude and distribution. Given the data, we therefore found insufficient evidence to reject the null hypothesis.

The Bayesian information criterion, on the other hand, strongly favored the Euclidean embedding in both cases. Due to the similar predictions, the difference in BIC scores are largely a result of the different number of free parameters needed to fully specify the models. In this sense, the metric constraints of the Euclidean embedding are advantageous, leading to a simpler model. In the non-metric labeled graph, each label is independent of other labels and must therefore be fully specified. Still, due to the non-Euclidean property of the wormhole environment, a perfect Euclidean embedding cannot exist and a difference between the models must remain. It is therefore surprising that the lack of metric constraints in the non-metric labeled graph did not lead to significantly smaller prediction errors.

In the original study, subjects explored the environment by walking continuous paths and thereby obtained information not only about the place-to-place distances and turns but also about the overall connectivity of the network. The conclusions drawn in Warren et al. (2017) [35] imply that this network information is not used for the shortcut task, which is thought to be solved by vector addition along the direct path only. Here, we showed that the behavioral data are also consistent with the idea of consolidating both distance and network information in a metrically embedded graph. We thus refute the conclusion in Warren et al. (2017) [35]: it is not necessary to discard Euclidean metric properties and to reduce the representation to a non-metric framework in order to explain the observed behavior.

The main difference between the vector navigation in the labeled graph and the embedded graph suggested here lies in the treatment of repeated distance and angle measurements during prolonged navigation. Repeated measurements might simply be used to improve the estimates of distances and angles for individual labels without exploiting the constraints that these measurements impose on adjacent labels and indeed on the entire graph. Metric embedding, in contrast, allows to make use of these constraints such that improved estimates of one edge will lead to better distance and angle estimates everywhere. In this view, the main advantage of having a metrically embedded representation of space is not so much its resemblance to a geographic map, but the possibility to integrate local and repeated measurements into a consolidated structure. The result is still a graph, but with metrically embedded vertices from which directions can be derived directly without the "mental path integration" procedure suggested by Warren et al. (2017) [35]. This finding is also strongly supported by the large difference in BIC score between the models arising from the difference in required parameters to specify them.

Note that even an optimal metric embedding is not necessarily equivalent to the Euclidean ground truth; cognitive space is not natural space, and the internal representation may still be systematically distorted, even under normal Euclidean circumstances. This might explain poor navigational performance even after prolonged exposure to the environment (e.g. [21]). Importantly, an internal Euclidean representation also does not preclude the possibility of biased *inference* about the world from that map. For example, it has long been known that judgments about distance and directions between places are biased by context and asymmetric [26, 46], which is in principle incompatible with a Euclidean metric map. However, it is possible to arrive at biased estimations if the estimation function itself is biased and context-dependent, even if the representation is not [46]. Here, distances and angles are explicitly not stored in the metric Euclidean representation and have to be inferred, which leaves room for such biases. Of course, it is difficult to disentangle factors caused by the representation from factors caused by the processing [26], but this is also true for judgments derived from the labeled graph model, and there is room for compromise:

Non-metric topological and metrically embedded information may also coexist. Combined models have previously been proposed, for example for different levels of spatial hierarchy [47, 48], where the local Euclidean structure of individual places or regions is known but higher-level relations between different regions are encoded as a graph. For example, a local plaza may be well-represented by a Euclidean metric map, but directions to other places within the city may only be memorized as a sequence of turns. In the context of this present study, this relates to the problem of what constitutes a vertex of the graph. In our simulation, we placed vertices at all corners of the maze, but other choices are possible. A neural network model assuming metric representations within small regions and categorical knowledge of these regions themselves has been presented by Baumann and Mallot (2023) [49].

Topological and metric information may also be used under different environmental constraints or at different stages of exploration and familiarization [17]. Initially, the environment may be encoded in terms of adjacency relations and individual routes, which then over time is consolidated in an encompassing map as the amount of information increases. This scenario is supported by reports that grid cell firing fields are initially anchored by the walls of individual compartments, but with experience extend across boundaries to encompass a larger space [50, 51]. The embedding algorithm presented here may also be considered a support, because it describes a transformation of local position information under topological constraints into a Euclidean metric map.

## Author Contributions

**Conceptualization:** Tristan Baumann, Hanspeter A. Mallot.

**Formal analysis:** Tristan Baumann.

**Software:** Tristan Baumann.

**Writing – original draft:** Tristan Baumann, Hanspeter A. Mallot.

**Writing – review & editing:** Tristan Baumann, Hanspeter A. Mallot.

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
