## [Decision Letter · Decision Letter 0]

21 Oct 2023

Dear Mr. Baumann,

Thank you very much for submitting your manuscript "Metric information in cognitive maps: Euclidean embedding of non-Euclidean environments" for consideration at PLOS Computational Biology. As with all papers reviewed by the journal, your manuscript was reviewed by members of the editorial board and by several independent reviewers. The reviewers appreciated the attention to an important topic. Based on the reviews, we are likely to accept this manuscript for publication, providing that you modify the manuscript according to the review recommendations.

In particular, the authors should make use of Bayesian statistics or information criteria for model selection (e.g. AIC or BIC) to make a more formal comparison between their embedding model and a topological framework; and - if possible - attempt to relate their embedding model to a wider range of published data sets, to ensure that their conclusions are robust. 

Sincerely,

Daniel Bush

Academic Editor

PLOS Computational Biology

Thomas Serre

Section Editor

PLOS Computational Biology

Reviewer's Responses to Questions

**Comments to the Authors:**

Reviewer #1: In this paper the authors propose a model that can account for deviation from Euclidean space in cognitive maps holding on the hypothesis that cognitive maps are fundamentally Euclidean. Basically, they suggest that, instead of learning and representing the position of objects/events in space using a non-euclidean graphical structure, what people usually do is to embed a non-metric model in 2D Euclidean coordinates. By re-analyzing the data from a previous behavioral experiment, they show that the embedded model and the graph model (non Euclidean) explain the data equally well. Although the embedded model, thus, is not necessarily better than a graph model, the former is preferable because it allows to integrate local repeated measures into a global structure (whereas in a non-metric graph-like cognitive map, each edge is independent from the others).

I liked the paper, it investigates a very important as well as difficult question: Whether humans navigate the environment constructing an Euclidean map. The author suggest that this is the case, proposing a hybrid model in which a non-Euclidean graph is embedded in a 2d Euclidean coordinates and re-analyzing an important dataset that, putatively, provide evidence for non-Euclidean navigation. I find the paper well written and, to the level of understanding that I can provide, which does not cover all the mathematical details as well as all the relevant literature on this specific argument (cognitive maps vs cognitive graphs), it seems solid: I could not find any specific problem in the analysis or the construction of the model.

Although, at the end, the paper cannot adjudicate between the labeled graph and the embedded graph model, I think it will be a good addition to the literature and the current debate. But I have some suggestions:

(1) The author can report Bayesian statistics to support the lack of difference between models

(2) Is there another dataset on which the two models can be compared? It would be more convincing to see a replication of the results using different data (with different non-Euclidean environments)

(3) P. 16 “The embedding may possibly be somewhat better at predicting the within-subject angular deviation in the rips and folds dataset, but the results did not pass the selected significance threshold” – I suggest to delete this sentence from the manuscript. The difference is not significant and it should not be commented as a hint of a possible difference.

Reviewer #2: The study utilized data from Warren et al. (2017) to investigate whether embedding a graph into Euclidean coordinates can explain data from a wormhole experiment better than a labeled graph. They used a numerical optimization method to derive embeddings, which are representations of the cognitive map. For their primary dataset, they derived shortcut predictions from both the non-metric labeled and embedded graph models. These predictions were then compared to human shortcut estimates from the Warren et al. (2017) study. The authors found that both the Euclidean embedding and the non-metric model predicted the data with similar accuracy. They conclude that since the embedding graph is simpler, it is a better model for the cognitive map than the non-metric models.

The authors employed a unique approach by using a numerical optimization method to derive embeddings, and the use of both non-metric labeled and embedded graph models to predict human behavior in navigating non-Euclidean environments is a solid approach. This approach takes the question about non-metric graphs into a new phase of direct comparisons between models. Given that most of the data were derived from another paper, design questions are minimal. I do have a few concerns that are important to address.

Major concerns:

The biggest concern is about the model comparison. The two models predicted the data with similar accuracy, which could lead to ambiguity in comparing the two models. The authors assume that the Euclidean embedded model is simpler, but there is not really much to support that claim. Conducing AIC or BIC to more directly compare the models would help, or at least providing more justification as to why metric embedding is simpler. A topological graph seems like a fairly simple idea to me. The metric embedding yields multi-level paths (e.g., Figures 4 and 5), which seems like it could be far more complex than the topological graph, and is also non-Euclidean.

Is the stress function simpler than a set of heuristics, such as regularizing to 90 degrees?

How does the embedding explain inconsistencies in regular Euclidean judgments, e.g., Diwadkar & McNamara, Tversky 1992, etc.? If the metric embedding simplifies to Euclidean structure in normal environments (see comment below), then how can it explain other violations of Euclidean geometry observed in the literature?

Minor concerns:

Metric embedding is central to this paper, but I didn’t see a really clear definition that could be operationalized (or tested/falsified in the future). It starts to come out on line 202, but an earlier definition would allow the reader to follow the arguments in the introduction. Similarly, details about the non-metric labeled graph model only come out on page 12. Much of those details are in the other paper, but a brief description (e.g., averaging of paths found by vector addition) would be useful.

When the authors say the embedding is not a simple averaging and so it won’t end up in the middle, it is not clear how that calculation is made to get the new vector.

How accurate are the measurements of the distances and angles for each triplet (e.g. lines 265)? If they are all very accurate, wouldn’t that lead to a Euclidean structure? Or is that the case if you are dealing with a metric map, but not with a wormhole? I think this might have been brought up later on but was not completely clear to me.

Abstract line 20 “so” should probably be “so-called”?

**Have the authors made all data and (if applicable) computational code underlying the findings in their manuscript fully available?**

Reviewer #1: Yes

Reviewer #2: Yes

PLOS authors have the option to publish the peer review history of their article (what does this mean?). If published, this will include your full peer review and any attached files.

Reviewer #1: No

Reviewer #2: No

Figure Files:

Data Requirements:

Reproducibility:

References:

---

## [Decision Letter · Decision Letter 1]

11 Dec 2023

Dear Mr. Baumann,

We are pleased to inform you that your manuscript 'Metric information in cognitive maps: Euclidean embedding of non-Euclidean environments' has been provisionally accepted for publication in PLOS Computational Biology.

Best regards,

Daniel Bush

Academic Editor

PLOS Computational Biology

Thomas Serre

Section Editor

PLOS Computational Biology

Reviewer's Responses to Questions

**Comments to the Authors:**

Reviewer #1: The authors replies are convincing and I think this paper will be a valuable contribution to the debate.

Reviewer #2: The authors have responded well to all of my comments.

**Have the authors made all data and (if applicable) computational code underlying the findings in their manuscript fully available?**

Reviewer #1: Yes

Reviewer #2: None

PLOS authors have the option to publish the peer review history of their article (what does this mean?). If published, this will include your full peer review and any attached files.

Reviewer #1: No

Reviewer #2: No

---

## [Editor Report · Acceptance letter]

20 Dec 2023

PCOMPBIOL-D-23-01146R1 

Metric information in cognitive maps: Euclidean embedding of non-Euclidean environments

Dear Dr Baumann,

I am pleased to inform you that your manuscript has been formally accepted for publication in PLOS Computational Biology. Your manuscript is now with our production department and you will be notified of the publication date in due course.

With kind regards,

Timea Kemeri-Szekernyes
